# Dynamic Importance Sampling for Anytime Bounds of the Partition Function

**Qi Lou**
Computer Science
Univ. of California, Irvine
Irvine, CA 92697, USA
qlou@ics.uci.edu

**Rina Dechter**
Computer Science
Univ. of California, Irvine
Irvine, CA 92697, USA
dechter@ics.uci.edu

**Alexander Ihler**
Computer Science
Univ. of California, Irvine
Irvine, CA 92697, USA
ihler@ics.uci.edu

## Abstract

Computing the partition function is a key inference task in many graphical models. In this paper, we propose a dynamic importance sampling scheme that provides anytime finite-sample bounds for the partition function. Our algorithm balances the advantages of the three major inference strategies, heuristic search, variational bounds, and Monte Carlo methods, blending sampling with search to refine a variationally defined proposal. Our algorithm combines and generalizes recent work on anytime search [16] and probabilistic bounds [15] of the partition function. By using an intelligently chosen weighted average over the samples, we construct an unbiased estimator of the partition function with strong finite-sample confidence intervals that inherit both the rapid early improvement rate of sampling and the long-term benefits of an improved proposal from search. This gives significantly improved anytime behavior, and more flexible trade-offs between memory, time, and solution quality. We demonstrate the effectiveness of our approach empirically on real-world problem instances taken from recent UAI competitions.

## 1 Introduction

Probabilistic graphical models, including Bayesian networks and Markov random fields, provide a framework for representing and reasoning with probabilistic and deterministic information [5, 6, 8]. Reasoning in a graphical model often requires computing the partition function, or normalizing constant of the underlying distribution. Exact computation of the partition function is known to be #P-hard [19] in general, leading to the development of many approximate schemes. Two important properties for a good approximation are that (1) it provides bounds or confidence guarantees on the result, so that the degree of approximation can be measured; and that (2) it can be improved in an anytime manner, so that the approximation becomes better as more computation is available.

In general, there are three major paradigms for approximate inference: variational bounds, heuristic search, and Monte Carlo sampling. Each method has advantages and disadvantages. Variational bounds [21], and closely related approximate elimination methods [7, 14] provide deterministic guarantees on the partition function. However, these bounds are not anytime; their quality often depends on the amount of memory available, and do not improve without additional memory. Search algorithms [12, 20, 16] explicitly enumerate over the space of configurations and eventually provide an exact answer; however, while some problems are well-suited to search, others only improve their quality very slowly with more computation. Importance sampling [e.g., 4, 15] gives probabilistic bounds that improve with more samples at a predictable rate; in practice this means bounds that improve rapidly at first, but are slow to become very tight. Several algorithms combine two strategies: approximate hash-based counting combines sampling (of hash functions) with CSP-based search [e.g., 3, 2] or other MAP queries [e.g., 9, 10], although these are not typically formulated to provide anytime

behavior. Most closely related to this work are [16] and [15], which perform search and sampling, respectively, guided by variational bounds.

In this work, we propose a dynamic importance sampling algorithm that provides anytime probabilistic bounds (i.e., they hold with probability $1 - \delta$ for some confidence parameter $\delta$). Our algorithm interleaves importance sampling with best first search [16], which is used to refine the proposal distribution of successive samples. In practice, our algorithm enjoys both the rapid bound improvement characteristic of importance sampling [15], while also benefiting significantly from search on problems where search is relatively effective, or when given enough computational resources, even when these points are not known in advance. Since our samples are drawn from a sequence of different, improving proposals, we devise a weighted average estimator that upweights higher-quality samples, giving excellent anytime behavior.

**Motivating example.** We illustrate the focus and contributions of our work on an example problem instance (Fig. 1). Search [16] provides strict bounds (gray) but may not improve rapidly, particularly once memory is exhausted; on the other hand, importance sampling [15] provides probabilistic bounds (green) that improve at a predictable rate, but require more and more samples to become tight. We first describe a "two stage" sampling process that uses a search tree to improve the baseline bound from which importance sampling starts (blue), greatly improving its long-term performance, then present our dynamic importance sampling (DIS) algorithm, which interleaves the search and sampling processes (sampling from a sequence of proposal distributions) to give bounds that are strong in an anytime sense.

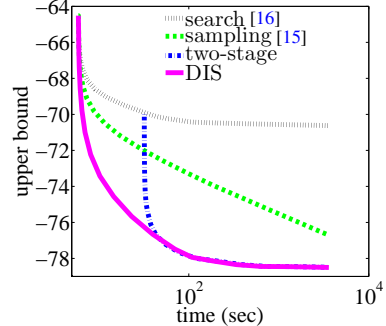

Figure 1: Example: bounds on $\log Z$ for `protein` instance 1bgc.

## 2 Background

Let $X = (X_1, \ldots, X_M)$ be a vector of random variables, where each $X_i$ takes values in a discrete domain $\mathcal{X}_i$; we use lower case letters, e.g. $x_i \in \mathcal{X}_i$, to indicate a value of $X_i$, and $x$ to indicate an assignment of $X$. A graphical model over $X$ consists of a set of factors $\mathcal{F} = \{f_\alpha(X_\alpha) \mid \alpha \in \mathcal{I}\}$, where each factor $f_\alpha$ is defined on a subset $X_\alpha = \{X_i \mid i \in \alpha\}$ of $X$, called its scope.

We associate an undirected graph $\mathcal{G} = (V, E)$ with $\mathcal{F}$, where each node $i \in V$ corresponds to a variable $X_i$ and we connect two nodes, $(i, j) \in E$, iff $\{i, j\} \subseteq \alpha$ for some $\alpha$. The set $\mathcal{I}$ then corresponds to cliques of $\mathcal{G}$. We can interpret $\mathcal{F}$ as an unnormalized probability measure, so that

$$f(x) = \prod_{\alpha \in \mathcal{I}} f_\alpha(x_\alpha), \qquad\qquad Z = \sum_x \prod_{\alpha \in \mathcal{I}} f_\alpha(x_\alpha)$$

$Z$ is called the *partition function*, and normalizes $f(x)$. Computing $Z$ is often a key task in evaluating the probability of observed data, model selection, or computing predictive probabilities.

### 2.1 AND/OR search trees

We first require some notations from search. AND/OR search trees are able to exploit the conditional independence properties of the model, as expressed by a *pseudo tree*:

**Definition 1** (pseudo tree)**.** *A pseudo tree of an undirected graph $\mathcal{G} = (V, E)$ is a directed tree $\mathfrak{T} = (V, E')$ sharing the same set of nodes as $\mathcal{G}$. The tree edges $E'$ form a subset of $E$, and we require that each edge $(i, j) \in E \setminus E'$ be a "back edge", i.e., the path from the root of $\mathfrak{T}$ to $j$ passes through $i$ (denoted $i \leq j$). $\mathcal{G}$ is called the primal graph of $\mathfrak{T}$.*

Fig. 2(a)-(b) show an example primal graph and pseudo tree. Guided by the pseudo tree, we can construct an AND/OR search tree $\mathcal{T}$ consisting of alternating levels of OR and AND nodes. Each OR node $s$ is associated with a variable, which we slightly abuse notation to denote $X_s$; the children of $s$, $ch(s)$, are AND nodes corresponding to the possible values of $X_s$. The root $\emptyset$ of the AND/OR search tree corresponds to the root of the pseudo tree. Let $pa(c) = s$ indicate the parent of $c$, and $an(c) = \{n \mid n \leq c\}$ be the ancestors of $c$ (including itself) in the tree.

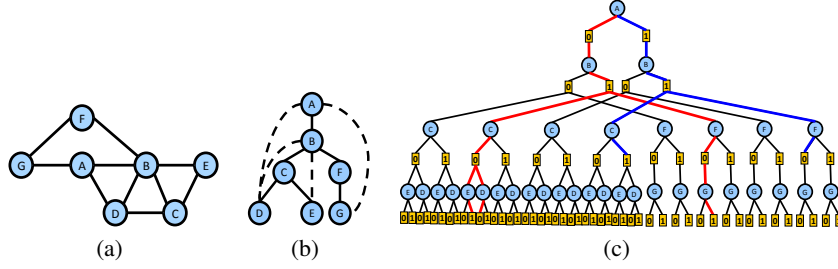

Figure 2: (a) A primal graph of a graphical model over 7 variables. (b) A pseudo tree for the primal graph consistent with elimination order $G, F, E, D, C, B, A$. (c) AND/OR search tree guided by the pseudo tree. One full solution tree is marked red and one partial solution tree is marked blue.

As the pseudo tree defines a partial ordering on the variables $X_i$, the AND/OR tree extends this to one over partial configurations of $X$. Specifically, any AND node $c$ corresponds to a partial configuration $x_{\leq c}$ of $X$, defined by its assignment and that of its ancestors: $x_{\leq c} = x_{\leq p} \cup \{X_s = x_c\}$, where $s = pa(c)$, $p = pa(s)$. For completeness, we also define $x_{\leq s}$ for any OR node $s$, which is the same as that of its AND parent, i.e., $x_{\leq s} = x_{\leq pa(s)}$. For any node $n$, the corresponding variables of $x_{\leq n}$ is denoted as $X_{\leq n}$. Let $de(X_n)$ be the set of variables below $X_n$ in the pseudo tree; we define $X_{>n} = de(X_n)$ if $n$ is an AND node; $X_{>n} = de(X_n) \cup \{X_n\}$ if $n$ is an OR node.

The notion of a *partial solution tree* captures partial configurations of $X$ respecting the search order:

**Definition 2** (partial solution tree). *A partial solution tree $T$ of an AND/OR search tree $\mathcal{T}$ is a subtree satisfying three conditions: (1) $T$ contains the root of $\mathcal{T}$; (2) if an OR node is in $T$, **at most** one of its children is in $T$; (3) if an AND node is in $T$, all of its children **or** none of its children are in $T$.*

Any partial solution tree $T$ defines a partial configuration $x_T$ of $X$; if $x_T$ is a complete configuration of $X$, we call $T$ a *full solution tree*, and use $T_x$ to denote the corresponding solution tree of a complete assignment $x$. Fig. 2(c) illustrates these concepts.

We also associate a weight $w_c$ with each AND node, defined to be the product of all factors $f_\alpha$ that are instantiated at $c$ but not before:

$$w_c = \prod_{\alpha \in \mathcal{I}_c} f_\alpha(x_\alpha), \qquad\qquad \mathcal{I}_c = \{\alpha \mid X_c \in X_\alpha \subseteq X_{\leq c}\}$$

For completeness, define $w_s = 1$ for any OR node $s$. It is then easy to see that, for any node $n$, the product of weights on a path to the root, $g_n = \prod_{a \leq n} w_a$ (termed the *cost* of the path), equals the value of the factors whose scope is fully instantiated at $n$, i.e., fully instantiated by $x_{\leq n}$. We can extend this cost notion to any partial solution tree $T$ by defining $g(T)$ as the product of all factors fully instantiated by $x_T$; we will slightly abuse notation by using $g(T)$ and $g(x_T)$ interchangeably. Particularly, the cost of any full solution tree equals the value of its corresponding complete configuration. We use $g(x_{>n}|x_{\leq n})$ (termed the *conditional cost*) to denote the quotient $g([x_{\leq n}, x_{>n}])/g(x_{\leq n})$, where $x_{>n}$ is any assignment of $X_{>n}$, the variables below $n$ in the search tree.

We give a "value" $v_n$ to each node $n$ equal to the total conditional cost of all configurations below $n$:

$$v_n = \sum_{x_{>n}} g(x_{>n}|x_{\leq n}). \tag{1}$$

The value of the root is simply the partition function, $v_\emptyset = Z$. Equivalently, $v_n$ can be defined recursively: if $n$ is an AND node corresponding to a leaf of the pseudo tree, let $v_n = 1$; otherwise,

$$v_n = \begin{cases} \prod_{c \in ch(n)} v_c, & \text{if AND node } n \\ \sum_{c \in ch(n)} w_c v_c, & \text{if OR node } n \end{cases} \tag{2}$$

## 2.2 AND/OR best-first search for bounding the partition function

AND/OR best-first search (AOBFS) can be used to bound the partition function in an anytime fashion by expanding and updating bounds defined on the search tree [16]. Beginning with only the root

$\emptyset$, AOBFS expands the search tree in a best-first manner. More precisely, it maintains an explicit AND/OR search tree of visited nodes, denoted $\mathcal{S}$. For each node $n$ in the AND/OR search tree, AOBFS maintains $u_n$, an upper bound on $v_n$, initialized via a pre-compiled heuristic $v_n \leq h_n^+$, and subsequently updated during search using information propagated from the frontier:

$$u_n = \begin{cases} \prod_{c \in ch(n)} u_c, & \text{if AND node } n \\ \sum_{c \in ch(n)} w_c u_c, & \text{if OR node } n \end{cases} \tag{3}$$

Thus, the upper bound at the root, $u_\emptyset$, is an anytime deterministic upper bound of $Z$. Note that this upper bound depends on the current search tree $\mathcal{S}$, so we write $U^\mathcal{S} = u_\emptyset$.

If all nodes below $n$ have been visited, then $u_n = v_n$; we call $n$ *solved* and can remove the subtree below $n$ from memory. Hence we can partition the frontier nodes into two sets: solved frontier nodes, *SOLVED*($\mathcal{S}$), and unsolved ones, *OPEN*($\mathcal{S}$). AOBFS assigns a priority to each node and expands a top-priority (unsolved) frontier node at each iteration. We use the "upper priority" from [16],

$$U_n = g_n u_n \prod_{s \in branch(n)} u_s \tag{4}$$

where $branch(n)$ are the OR nodes that are siblings of some node $\leq n$. $U_n$ quantifies $n$'s contribution to the global bound $U^\mathcal{S}$, so this priority attempts to reduce the upper bound on $Z$ as quickly as possible.

We can also interpret our bound $U^\mathcal{S}$ as a sum of bounds on each of the partial configurations covered by $\mathcal{S}$. Concretely, let $\mathbb{T}^\mathcal{S}$ be the set of projections of full solution trees on $\mathcal{S}$ (in other words, $\mathbb{T}^\mathcal{S}$ are partial solution trees whose leaves are frontier nodes of $\mathcal{S}$); then,

$$U^\mathcal{S} = \sum_{T \in \mathbb{T}^\mathcal{S}} U_T \qquad \text{where} \qquad U_T = g(T) \prod_{s \in leaf(T)} u_s \tag{5}$$

and $leaf(T)$ are the leaf nodes of the partial solution tree $T$.

## 2.3 Weighted mini-bucket for heuristics and sampling

To construct a heuristic function for search, we can use a class of variational bounds called *weighted mini-bucket* (WMB, [14]). WMB corresponds to a relaxed variable elimination procedure, respecting the search pseudo tree order, that can be tightened using reparameterization (or "cost-shifting") operations. Importantly for this work, this same relaxation can also be used to define a proposal distribution for importance sampling that yields finite-sample bounds [15]. We describe both properties here.

Let $n$ be any node in the search tree; then, one can show that WMB yields the following reparametrization of the conditional cost below $n$ [13]:

$$g(x_{>n}|x_{\leq n}) = h_n^+ \prod_k \prod_j b_{kj}(x_k|x_{an_j(k)})^{\rho_{kj}}, \quad X_k \in X_{>n} \tag{6}$$

where $X_{an_j(k)}$ are the ancestors of $X_k$ in the pseudo tree that are included in the $j$-th mini-bucket of $X_k$. The size of $X_{an_j(k)}$ is controlled by a user-specified parameter called the *ibound*. The $b_{kj}(x_k|x_{an_j(k)})$ are conditional beliefs, and the non-negative weights $\rho_{kj}$ satisfy $\sum_j \rho_{kj} = 1$.

Suppose that we define a conditional distribution $q(x_{>n}|x_{\leq n})$ by replacing the geometric mean over the $b_{kj}$ in (6) with their arithmetic mean:

$$q(x_{>n}|x_{\leq n}) = \prod_k \sum_j \rho_{kj} b_{kj}(x_k|x_{an_j(k)}) \tag{7}$$

Applying the arithmetic-geometric mean inequality, we see that $g(x_{>n}|x_{\leq n})/h_n^+ \leq q(x_{>n}|x_{\leq n})$. Summing over $x_{>n}$ shows that $h_n^+$ is a valid upper bound heuristic for $v_n$:

$$v_n = \sum_{x_{>n}} g(x_{>n}|x_{\leq n}) \leq h_n^+$$

The mixture distribution $q$ can be also used as a proposal for importance sampling, by drawing samples from $q$ and averaging the importance weights, $g/q$. For any node $n$, we have that

$$g(x_{>n}|x_{\leq n})/q(x_{>n}|x_{\leq n}) \leq h_n^+, \qquad \mathbb{E}\left[g(x_{>n}|x_{\leq n})/q(x_{>n}|x_{\leq n})\right] = v_n \tag{8}$$

i.e., the importance weight $g(x_{>n}|x_{\leq n})/q(x_{>n}|x_{\leq n})$ is an unbiased and bounded estimator of $v_n$.

In [15], this property was used to give finite-sample bounds on $Z$ which depended on the WMB bound, $h_\emptyset^+$. To be more specific, note that $g(x_{>n}|x_{\leq n}) = f(x)$ when $n$ is the root $\emptyset$, and thus $f(x)/q(x) \leq h_\emptyset^+$; the boundedness of $f(x)/q(x)$ results in the following finite-sample upper bound on $Z$ that holds with probability at least $1 - \delta$:

$$Z \leq \frac{1}{N} \sum_{i=1}^{N} \frac{f(x^i)}{q(x^i)} + \sqrt{\frac{2\widehat{\mathrm{Var}}(\{f(x^i)/q(x^i)\}_{i=1}^N) \ln(2/\delta)}{N}} + \frac{7 \ln(2/\delta) h_\emptyset^+}{3(N-1)} \qquad (9)$$

where $\{x^i\}_{i=1}^N$ are i.i.d. samples drawn from $q(x)$, and $\widehat{\mathrm{Var}}(\{f(x^i)/q(x^i)\}_{i=1}^N)$ is the unbiased empirical variance. This probabilistic upper bound usually becomes tighter than $h_\emptyset^+$ very quickly. A corresponding finite-sample lower bound on $Z$ exists as well [15].

## 3   Two-step sampling

The finite-sample bound (9) suggests that improvements to the upper bound on $Z$ may be translatable into improvements in the probabilistic, sampling bound. In particular, if we define a proposal that uses the search tree $\mathcal{S}$ and its bound $U^{\mathcal{S}}$, we can improve our sample-based bound as well. This motivates us to design a *two-step* sampling scheme that exploits the refined upper bound from search; it is a top-down procedure starting from the root:

**Step 1** For an internal node $n$: if it is an AND node, all its children are selected; if $n$ is an OR node, one child $c \in ch(n)$ is randomly selected with probability $w_c u_c / u_n$.

**Step 2** When a frontier node $n$ is reached, if it is unsolved, draw a sample of $X_{>n}$ from $q(x_{>n}|x_{\leq n})$; if it is solved, quit.

The behavior of **Step 1** can be understood by the following proposition:

**Proposition 1.** **Step 1** returns a partial solution tree $T \in \mathbb{T}^{\mathcal{S}}$ with probability $U_T / U^{\mathcal{S}}$ (see (5)). Any frontier node of $\mathcal{S}$ will be reached with probability proportional to its upper priority defined in (4).

Note that at **Step 2**, although the sampling process terminates when a solved node $n$ is reached, we associate every configuration $x_{>n}$ of $X_{>n}$ with probability $g(x_{>n}|x_{\leq n})/v_n$ which is appropriate in lieu of (1). Thus, we can show that this two-step sampling scheme induces a proposal distribution, denoted $q^{\mathcal{S}}(x)$, which can be expressed as:

$$q^{\mathcal{S}}(x) = \prod_{n \in AND(T_x \cap \mathcal{S})} w_n u_n / u_{pa(n)} \prod_{n' \in OPEN(\mathcal{S}) \cap T_x} q(x_{>n'}|x_{\leq n'}) \prod_{n'' \in SOLVED(\mathcal{S}) \cap T_x} g(x_{>n''}|x_{\leq n''}) / v_{n''}$$

where $AND(T_x \cap \mathcal{S})$ is the set of all AND nodes of the partial solution tree $T_x \cap \mathcal{S}$. By applying (3), and noticing that the upper bound is the initial heuristic for any node in $OPEN(\mathcal{S})$ and is exact at any solved node, we re-write $q^{\mathcal{S}}(x)$ as

$$q^{\mathcal{S}}(x) = \frac{g(T_x \cap \mathcal{S})}{U^{\mathcal{S}}} \prod_{n' \in OPEN(\mathcal{S}) \cap T_x} h_{n'}^+ \, q(x_{>n'}|x_{\leq n'}) \prod_{n'' \in SOLVED(\mathcal{S}) \cap T_x} g(x_{>n''}|x_{\leq n''}) \qquad (10)$$

$q^{\mathcal{S}}(x)$ actually provides bounded importance weights that can use the refined upper bound $U^{\mathcal{S}}$:

**Proposition 2.** Importance weights from $q^{\mathcal{S}}(x)$ are bounded by the upper bound of $\mathcal{S}$, and are unbiased estimators of $Z$, i.e.,

$$f(x)/q^{\mathcal{S}}(x) \leq U^{\mathcal{S}}, \qquad\qquad \mathbb{E}\left[f(x)/q^{\mathcal{S}}(x)\right] = Z \qquad (11)$$

*Proof.* Note that $f(x)$ can be written as

$$f(x) = g(T_x \cap \mathcal{S}) \prod_{n' \in OPEN(\mathcal{S}) \cap T_x} g(x_{>n'}|x_{\leq n'}) \prod_{n'' \in SOLVED(\mathcal{S}) \cap T_x} g(x_{>n''}|x_{\leq n''}) \qquad (12)$$

Noticing that for any $n' \in OPEN(\mathcal{S})$, $g(x_{>n'}|x_{\leq n'}) \leq h_{n'}^+ \, q(x_{>n'}|x_{\leq n'})$ by (8), and comparing with (10), we see $f(x)/q^{\mathcal{S}}(x)$ is bounded by $U^{\mathcal{S}}$. Its unbiasedness is trivial. $\qquad\square$

---

**Algorithm 1** Dynamic importance sampling (DIS)

---

**Require:** Control parameters $N_d$, $N_l$; memory budget, time budget.
**Ensure:** $N$, HM($\boldsymbol{U}$), $\widehat{\text{Var}}(\{\widehat{Z}_i/U_i\}_{i=1}^N)$, $\widehat{Z}$, $\Delta$.
 1: Initialize $\mathcal{S} \leftarrow \{\emptyset\}$ with the root $\emptyset$.
 2: **while** within the time budget
 3:     **if** within the memory budget                *// update $\mathcal{S}$ and its associated upper bound $U^{\mathcal{S}}$*
 4:         Expand $N_d$ nodes via AOBFS (Alg. 1 of [16]) with the upper priority defined in (4).
 5:     **end if**
 6:     Draw $N_l$ samples via TWOSTEPSAMPLING($\mathcal{S}$).
 7:     After drawing each sample:
 8:         Update $N$, HM($\boldsymbol{U}$), $\widehat{\text{Var}}(\{\widehat{Z}_i/U_i\}_{i=1}^N)$.
 9:         Update $\widehat{Z}$, $\Delta$ via (13), (14).
10: **end while**

11: **function** TWOSTEPSAMPLING($\mathcal{S}$)
12:     Start from the root of the search tree $\mathcal{S}$:
13:         For an internal node $n$: select all its children if it is an AND node; select exactly
14:         one child $c \in ch(n)$ with probability $w_c u_c / u_n$ if it is an OR node.
15:         At any unsolved frontier node $n$, draw one sample from $q(x_{>n}|x_{\leq n})$ in (7).
16: **end function**

---

Thus, importance weights resulting from our two-step sampling can enjoy the same type of bounds described in (9). Moreover, note that at any solved node, our sampling procedure incorporates the "exact" value of that node into the importance weights, which serves as Rao-Blackwellisation and can potentially reduce variance.

We can see that if $\mathcal{S} = \emptyset$ (before search), $q^{\mathcal{S}}(x)$ is the proposal distribution of [15]; as search proceeds, the quality of the proposal distribution improves (gradually approaching the underlying distribution $f(x)/Z$ as $\mathcal{S}$ approaches the complete search tree). If we perform search first, up to some memory limit, and then sample, which we refer to as *two-stage* sampling, our probabilistic bounds will proceed from an improved baseline, giving better bounds at moderate to long computation times. However, doing so sacrifices the quick improvement early on given by basic importance sampling. In the next section, we describe our dynamic importance sampling procedure, which balances these two properties.

## 4 Dynamic importance sampling

To provide good anytime behavior, we would like to do both sampling and search, so that early samples can improve the bound quickly, while later samples obtain the benefits of the search tree's improved proposal. To do so, we define a dynamic importance sampling (DIS) scheme, presented in Alg. 1, which interleaves drawing samples and expanding the search tree.

One complication of such an approach is that each sample comes from a different proposal distribution, and thus has a different bound value entering into the concentration inequality. Moreover, each sample is of a different quality – later samples should have lower variance, since they come from an improved proposal. To this end, we construct an estimator of $Z$ that upweights higher-quality samples. Let $\{x^i\}_{i=1}^N$ be a series of samples drawn via Alg. 1, with $\{\widehat{Z}_i = f(x^i)/q^{\mathcal{S}_i}(x^i)\}_{i=1}^N$ the corresponding importance weights, and $\{U_i = U^{\mathcal{S}_i}\}_{i=1}^N$ the corresponding upper bounds on the importance weights respectively. We introduce an estimator $\widehat{Z}$ of $Z$:

$$\widehat{Z} = \frac{\text{HM}(\boldsymbol{U})}{N} \sum_{i=1}^N \frac{\widehat{Z}_i}{U_i}, \qquad\qquad \text{HM}(\boldsymbol{U}) = \Big[\frac{1}{N} \sum_{i=1}^N \frac{1}{U_i}\Big]^{-1} \qquad (13)$$

where HM($\boldsymbol{U}$) is the harmonic mean of the upper bounds $U_i$. $\widehat{Z}$ is an unbiased estimator of $Z$ (since it is a weighted average of independent, unbiased estimators). Additionally, since $Z/\,\text{HM}(\boldsymbol{U})$, $\widehat{Z}/\,\text{HM}(\boldsymbol{U})$, and $\widehat{Z}_i/U_i$ are all within the interval $[0,1]$, we can apply an empirical Bernstein bound [17] to derive finite-sample bounds:

**Theorem 1.** Define the deviation term

$$\Delta = \text{HM}(\boldsymbol{U})\Big(\sqrt{\frac{2\widehat{\text{Var}}(\{\widehat{Z}_i/U_i\}_{i=1}^N)\ln(2/\delta)}{N}} + \frac{7\ln(2/\delta)}{3(N-1)}\Big) \quad (14)$$

where $\widehat{\text{Var}}(\{\widehat{Z}_i/U_i\}_{i=1}^N)$ is the unbiased empirical variance of $\{\widehat{Z}_i/U_i\}_{i=1}^N$. Then $\widehat{Z} + \Delta$ and $\widehat{Z} - \Delta$ are upper and lower bounds of $Z$ with probability at least $1 - \delta$, respectively, i.e., $\Pr[Z \leq \widehat{Z} + \Delta] \geq 1 - \delta$ and $\Pr[Z \geq \widehat{Z} - \Delta] \geq 1 - \delta$.

It is possible that $\widehat{Z} - \Delta < 0$ at first; if so, we may replace $\widehat{Z} - \Delta$ with any non-trivial lower bound of $Z$. In the experiments, we use $\widehat{Z}\delta$, a $(1 - \delta)$ probabilistic bound by the Markov inequality [11]. We can also replace $\widehat{Z} + \Delta$ with the current deterministic upper bound if the latter is tighter.

Intuitively, our DIS algorithm is similar to Monte Carlo tree search (MCTS) [1], which also grows an explicit search tree while sampling. However, in MCTS, the sampling procedure is used to grow the tree, while DIS uses a classic search priority. This ensures that the DIS samples are independent, since samples do not influence the proposal distribution of later samples. This also distinguishes DIS from methods such as adaptive importance sampling (AIS) [18].

# 5 Empirical evaluation

We evaluate our approach (DIS) against AOBFS (search, [16]) and WMB-IS (sampling, [15]) on several benchmarks of real-world problem instances from recent UAI competitions. Our benchmarks include `pedigree`, 22 genetic linkage instances from the UAI'08 inference challenge[1]; `protein`, 50 randomly selected instances made from the "small" protein side-chains of [22]; and BN, 50 randomly selected Bayesian networks from the UAI'06 competition[2]. These three sets are selected to illustrate different problem characteristics; for example `protein` instances are relatively small ($M = 100$ variables on average, and average induced width 11.2) but high cardinality (average $\max |\mathcal{X}_i| = 77.9$), while `pedigree` and BN have more variables and higher induced width (average $M$ 917.1 and 838.6, average width 25.5 and 32.8), but lower cardinality (average $\max |\mathcal{X}_i|$ 5.6 and 12.4).

We alloted 1GB memory to all methods, first computing the largest ibound that fits the memory budget, and using the remaining memory for search. All the algorithms used the same upper bound heuristics, which also means DIS and AOBFS had the same amount of memory available for search. For AOBFS, we use the memory-limited version (Alg. 2 of [16]) with "upper" priority, which continues improving its bounds past the memory limit. Additionally, we let AOBFS access a lower bound heuristic for no cost, to facilitate comparison between DIS and AOBFS. We show DIS for two settings, ($N_l$=1, $N_d$=1) and ($N_l$=1, $N_d$=10), balancing the effort between search and sampling. Note that WMB-IS can be viewed as DIS with ($N_l$=Inf, $N_d$=0), i.e., it runs pure sampling without any search, and two-stage sampling viewed as DIS with ($N_l$=1, $N_d$=Inf), i.e., it searches to the memory limit then samples. We set $\delta = 0.025$ and ran each algorithm for 1 hour. All implementations are in C/C++.

**Anytime bounds for individual instances.** Fig. 3 shows the anytime behavior of all methods on two instances from each benchmark. We observe that compared to WMB-IS, DIS provides better upper and lower bounds on all instances. In 3(d)–(f), WMB-IS is not able to produce tight bounds within 1 hour, but DIS quickly closes the gap. Compared to AOBFS, in 3(a)–(c),(e), DIS improves much faster, and in (d),(f) it remains nearly as fast as search. Note that four of these examples are sufficiently hard to be unsolved by a variable elimination-based exact solver, even with several orders of magnitude more computational resources (200GB memory, 24 hour time limit).

Thus, DIS provides excellent anytime behavior; in particular, ($N_l$=1, $N_d$=10) seems to work well, perhaps because expanding the search tree is slightly faster than drawing a sample (since the tree depth is less than the number of variables). On the other hand, two-stage sampling gives weaker early bounds, but is often excellent at longer time settings.

**Aggregated results across the benchmarks.** To quantify anytime performance of the methods in each benchmark, we introduce a measure based on the area between the upper and lower bound of

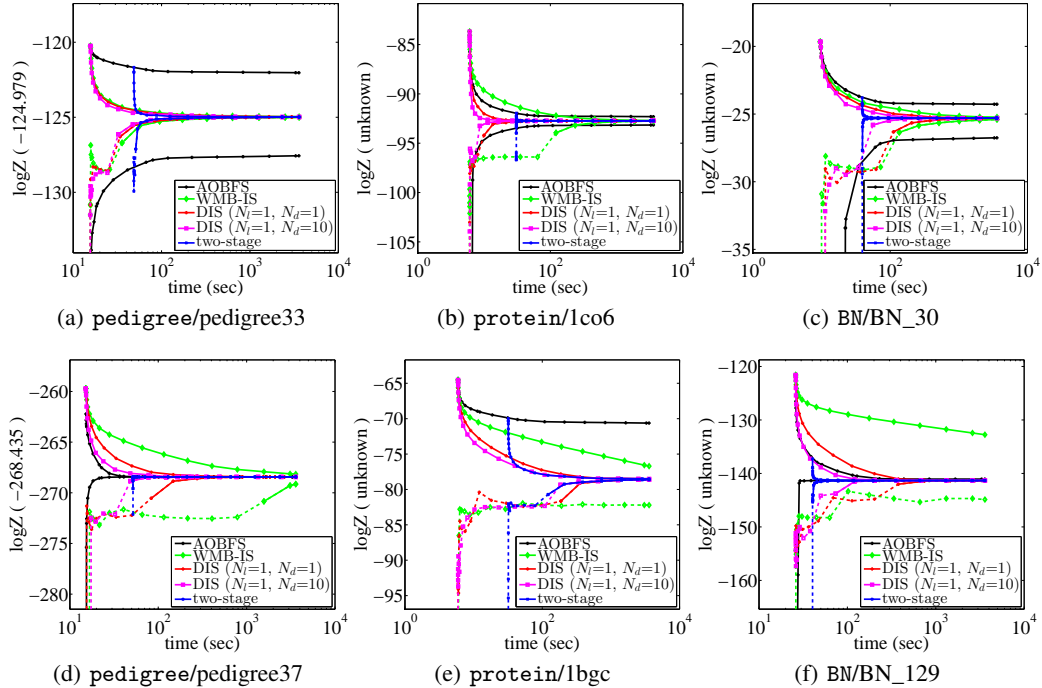

Figure 3: Anytime bounds on $\log Z$ for two instances per benchmark. Dotted line sections on some curves indicate Markov lower bounds. In examples where search is very effective (d,f), or where sampling is very effective (a), DIS is equal or nearly so, while in (b,c,e) DIS is better than either.

Table 1: Mean area between upper and lower bounds of $\log Z$, normalized by WMB-IS, for each benchmark. Smaller numbers indicate better anytime bounds. The best for each benchmark is bolded.

|  | AOBFS | WMB-IS | DIS ($N_l$=1, $N_d$=1) | DIS ($N_l$=1, $N_d$=10) | two-stage |
|---|---|---|---|---|---|
| pedigree | 16.638 | 1 | 0.711 | **0.585** | 1.321 |
| protein | 1.576 | 1 | 0.110 | **0.095** | 2.511 |
| BN | 0.233 | 1 | 0.340 | **0.162** | 0.865 |

$\log Z$. For each instance and method, we compute the area of the interval between the upper and lower bound of $\log Z$ for that instance and method. To avoid vacuous lower bounds, we provide each algorithm with an initial lower bound on $\log Z$ from WMB. To facilitate comparison, we normalize the area of each method by that of WMB-IS on each instance, then report the geometric mean of the normalized areas across each benchmark in Table 1. This shows the average relative quality compared to WMB-IS; smaller values indicate tighter anytime bounds. We see that on average, search is more effective than sampling on the BN instances, but much less effective on pedigree. Across all three benchmarks, DIS ($N_l$=1, $N_d$=10) produces the best result by a significant margin, while DIS ($N_l$=1, $N_d$=1) is also very competitive, and two-stage sampling does somewhat less well.

## 6 Conclusion

We propose a dynamic importance sampling algorithm that embraces the merits of best-first search and importance sampling to provide anytime finite-sample bounds for the partition function. The AOBFS search process improves the proposal distribution over time, while our particular weighted average of importance weights gives the resulting estimator quickly decaying finite-sample bounds, as illustrated on several UAI problem benchmarks. Our work also opens up several avenues for future research, including investigating different weighting schemes for the samples, more flexible balances between search and sampling (for example, changing over time), and more closely integrating the variational optimization process into the anytime behavior.

## Acknowledgements

We thank William Lam, Wei Ping, and all the reviewers for their helpful feedback.

This work is sponsored in part by NSF grants IIS-1526842, IIS-1254071, and by the United States Air Force under Contract No. FA8750-14-C-0011 and FA9453-16-C-0508.

## Footnotes

[1] http://graphmod.ics.uci.edu/uai08/Evaluation/Report/Benchmarks/

[2] http://melodi.ee.washington.edu/~bilmes/uai06InferenceEvaluation/

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
