[Reviews · NeurIPS 2017]

Reviewer 1



This paper presents a new anytime algorithm for estimating the value of the partition function. The proposed algorithm balances between search and sampling and it seems to give good empirical results. The paper is well-motivated and the proposed algorithm may have practical significance. The paper was quite technical and hard to follow. There was no high-level intuitive description of the procedure but it had to be inferred from the technical details. Algorithm 1 should probably have also other input such as the factors. I guess that in step 4, one expands S. In section 2.3, it is unclear what is a mini-bucket. What is h^+_N? On line 115 it seems to be an upper bound initialized using a precomputed heuristics but on line 147 it is the WMB bound.

Reviewer 2



The authors present a method for estimating the partition function that alternates between performing heuristic search and importance sampling. The estimated value of the partition function is confidence bounded and improves with additional computation time. Experimental evaluation is performed on problems from 2006 and 2008 UAI competitions by comparing confidence bounds of the proposed method against previous work that uses only sampling [15] or search [16]. The proposed method significantly outperforms sampling on certain problems and search on others, while maintaining performance roughly comparable to or better than either sampling or search across all problems. The originality of the work is a bit limited, as it is a fairly straightforward (but novel as far as I know) combination of two recent papers, references [15] and [16]. To review past work, [15] proposes a method that combines importance sampling with variational optimization. The authors chose a proposal distribution that is the solution to a variational optimization problem and show this choice can be interpreted as picking the proposal distribution that minimizes an upper bound on the value of importance weights. The importance weight bound is then used to construct a (probabilistic, but with confidence intervals) bound on the estimate of the partition function. [16] describes a search algorithm for computing deterministic bounds on the partition function. Unlike deterministic bounds given by variational methods, the search algorithm proposed in [16] can continue to improve its estimate in an anytime manner, even when given limited memory resources. This paper combines the methods of approaches of [15] and [16] by alternating between importance sampling and search. Search is performed using a weighted mini-bucket variational heuristic. The same mini-buckets are used to improve the proposal distribution over time, generating better samples. The authors provide a method for weighting later samples more heavily as the proposal distribution improves. The authors demonstrate that the proposed method improves estimation of the partition function over state of the art sampling and search based methods ([15] and [16]). There is room for researchers to build on this paper in the future, particularly in the areas of weighting samples generated by a set of proposal distributions that improve with time and exploring when computation should be focused more on sampling or search. The paper is clear and well organized. Previous work the paper builds on is well cited. Minor: - Line 243 states that DIS remains nearly as fast as AOBFS in figures 3(d) and (f), but it appears that the lower bound of DIS converges roughly half an order of magnitude slower. Line 233 states, "we let AOBFS access a lower bound heuristic for no cost." Does the lower bound converge more slowly for DIS than AOBFS in these instances or is the lower bound shown for AOBFS unfair? - Line 190, properites is a typo. - Line 146 states that, "In [15], this property was used to give finite-sample bounds on Z which depended on the WMB bound, h_{\emptyset}^{+}." It appears that in [15] the bound was only explicitly written in terms of Z_{trw}. Mentioning Z_{trw} may make it slightly easier to follow the derivation from [15].

Reviewer 3



This paper presents a method -- dynamic importance sampling (DIS) -- for computing bounds on the partition function of a probabilistic graphical model in an anytime fashion. DIS combines and extends two existing methods for estimating the partition function -- sampling and search -- which it interleaves in order to gain the benefits of each. A very well-written paper that clearly lays out its thesis, contributions, and results. The paper is technically sound, with claims that are well-supported both empirically and theoretically. The writing is clear and concise. The paper nicely introduces and builds on existing work before introducing its novel contributions, which look quite promising. My only comment is that I'd like to see empirical results on applications that seem less toy and where anytime estimation of (bounds on) the partition function is directly beneficial towards some final goal, such as cost reduction, accuracy improvement, etc.